# Black flies and Onchocerciasis: Knowledge, attitude and practices among inhabitants of Alabameta, Osun State, Southwestern, Nigeria

**Lateef O. Busari**[1]\*, **Monsuru Adebayo Adeleke**[1], **Olabanji A. Surakat**[1], **Akeem A. Akindele**[2], **Kamilu Ayo Fasasi**[1], **Olusola Ojurongbe**[2]

**1** Department of Zoology, Osun State University, Osogbo, Osun State, Nigeria, **2** Department of Medical Microbiology and Parasitology, Ladoke Akintola University of Technology, Ogbomosho, Oyo State, Nigeria

\* lateef.busari@pgc.uniosun.edu.ng

## Abstract

### Background and objectives

This study reports knowledge of residents of Alabameta community, Osun State, Nigeria on the bioecology and socio-economic burden of black flies and onchocerciasis.

### Methods

Using structured questionnaires and Focus Group Discussion (FGD), a total of 150 community respondents participated in the study.

### Results

The knowledge of the residents on the existence of black flies in the community was significant ($p < 0.05$) as all the 150 respondents confirmed the presence of black flies with the local name 'Amukuru' i.e causing itching. However, their lack of knowledge of the flies breeding site (104) (69%), prevention (134) (89%), cause (132) (88%), and treatment (133) (89%) of onchocerciasis was profound. Majority 147(98%) of the respondents reported that flies bite more in the wet season as against dry season 3(2%) and have a higher affinity (124) (82%) for biting the leg than any other part of the body. A larger percentage (89%) of the respondents are unaware of any medication for the treatment of onchocerciasis while 11% are aware. There had been no sensitization on onchocerciasis according to 89% of the respondents.

### Conclusion

Due to lack of resident's knowledge on black flies bioecology which may continuously expose them to the bite of the flies and ultimately infection, it is paramount that the Osun State government and the NTD implementing partner map out new public health education strategies during routine Mass Administration of Medicines with Ivermectin with a view to preventing onchocerciasis infection as well as man-vector contact.

**Data Availability Statement:** All relevant data are within the manuscript.

**Funding:** The author(s) received no specific funding for this work.

**Competing interests:** The authors have declared no competing interest exist.

## Author summary

The public health menace of parasites and their vectors is an issue of global public health concern. Black flies, the vector that transmit the dreadful parasite *Onchocerca volvulus* to its human host where it causes the debilitating disease, onchocerciasis, is a neglected tropical disease (NTD) with its greatest burden in sub-Saharan Africa. Nigeria is an endemic country for onchocerciasis. Despite efforts by health agencies globally in eradicating the disease through public health awareness, chemotherapy through community directed treatment with ivermectin (CDTI) and so on, it appears that inhabitants of Alabameta community, Osun State, southwestern, Nigeria are unaware of the bioecology of the vector as well as the public health implication of the disease according to the present study. Therefore, there is the urgent need for the government at the state and federal levels as well as health agencies to improve on public health awareness of the disease with a view to curbing exposure of residents and ultimately disease eradication.

## Introduction

Among the filarial nematodes, the public health significance of *Onchocerca volvulus* cannot be overemphasized being the causative organism of the dreadful and debilitating disease onchocerciasis [1]. *O. volvulus*, is transmitted by members of *Simulium damnosum* complex (black flies) through their bite while taking a blood meal. The flies breed in fast flowing, well–oxygenated rivers and streams where the female lay her egg and the larval and pupal stages of the vector develop [2]. Adult worms mates and produces tiny juvenile worms that migrate throughout the skin and eyes, causing the various symptoms of the disease [3,4].

The disease occurs mainly in tropical areas with more than 99% of infected people living in 31 countries in sub-Saharan Africa [5]. Human onchocerciasis (river blindness) causes severe itching and various skin changes with some infected people developing eye lesions that could lead to temporary or permanent blindness while nodules containing the adult worm [5].The disease is known to be endemic in many tropical countries with 18 million people infected worldwide and 120 million people at risk of the disease [6].

In Nigeria, the first case of onchocerciasis was reported by Parson in northern Nigeria in 1908 [7]. Nigeria ranks among countries that are endemic [8] with prevalence in all her states except Lagos, Katsina, Bayelsa and Rivers according to the Nigeria onchocerciasis elimination plan [2]. Studies have shown that six of the nine main West African members of the *Simulium damnosum* complex are present in Nigeria [9]. The species include *Simulium sirbanum*, *S. damnosum sensu stricto*, *Simulium sanctipauli*, *Simulium soubrense*, *Simulium squamosum*, *Simulium yahense*. The first two pairs are known as Savannah flies transmitting the savannah strain of *O. volvulus* while the remaining are forest flies transmitting the forest strain of the parasite causing more of skin diseases than blinding [10,11].

Osun State, a south-western state is mesoendemic for onchocerciasis with sixteen endemic Local Government areas (LGA) according to the rapid entomological mapping of onchocerciasis in Nigeria in 1995 and 1996. The state has a prevalence level of at least 20% and 40% onchocercal nodules and skin microfilarial infection respectively [12]. Although, there are reports on bioecology, larva ecology, species composition of black flies in the state. The reports have not been holistic, thus the need for further studies in other parts of the state with a view to understanding better the bioecology and socio-economic burden of both the vector and disease in order to assist in the effective planning of control methods in the state.

The present study therefore seeks to provide information on community knowledge, attitude and practice in Alabameta, Osun State, Nigeria on bioecology and socio-economic burden of black flies with a view to understanding its public health implication. This will help in planning effective control strategies in the study area and Osun State, Nigeria at large.

## Materials and methods

### Ethics statement

Ethical approval was sought and obtained from the Osun State University Health Research Ethics Committee. Verbal Consent of guardian of participants below age eighteen was sought and obtained.

Study Area: The study community is located in Ife South Local Government of Osun State, Southwest, Nigeria with an area of 730 km$^2$ and a population of 135,338(2006 census). Owena River transverses the community and extends to Ondo State. The river usually produces rapids along its course which serves as conducive breeding sites for *S. damnosum* [13].

Alabameta had been previously surveyed for onchocerciasis with a prevalence of 33.3% [14]. The intense biting of black flies had been reported in the community. This characteristics informed its selection for this study. The community lacks basic social amenities such hospital/clinic, electricity, pipe borne water, hospitals, good roads, and recreation center. Despite the high prevalence of onchocerciasis in the community, the community (nor neighboring communities) has not been enrolled on annual treatment of Mass Drug Administration. This may be partly due to its location as Alabameta is one of the hard to reach communities in Ife South Local Government of Osun State. The community population is about 300 of which majority are youths and school aged children.

Community awareness and mobilization was conducted prior to the commencement of the study through the village head and the head of the community secondary school to educate residents on the public health hazard of black flies and onchocerciasis.

Questionnaire Administration and Focus Group Discussion (FGD): The residents of the community were mobilized for the survey. A total of 150 structured questionnaires were administered to consented community members that are fifteen years and above to assess their knowledge on the bioecology and socioeconomic burden of the black fly. The questions captured the place of bite, season/month with the highest biting incidence and time, breeding site, effects of bite, body part mostly bitten and ways adopted to preventing fly bite. The questionnaires were complemented with Focus Group Discussion (FGD) of seven participants which included old men, old women and youths. The questions for the focus group discussion were unstructured and targeted at seeking information about the resident's general knowledge on the bioecology, attitudes and practice against black flies.

Data Analysis: The data from the study was subjected to T-test and Chi-square to determine the significant difference in dynamics and bioecology of black flies during the period of the study. All analysis was performed using SPSS version 17.

## Results

A total of 150 respondents were interviewed. Respondents within the age bracket 31–50 (69) (46%) were represented at a higher proportion than those within the age brackets 15–30 (54) (36%) and 51 above (27) (18%). There were more male respondents (76) (51%) than female respondents (74) (49%). About 51% (76) of the respondents were without formal education while 6% were educated up to tertiary level. Farming was the major occupation (48%), followed by trading (26%) and fishing (11%) (Table 1).

**Table 1. Demographic data of respondents of the community.**

| Characteristic | Age group | No. of respondent | Percentage % |
|---|---|---|---|
| Age | 15–30 | 54 | 36 |
| | 31–50 | 69 | 46 |
| | 51 above | 27 | 18 |
| Gender | Male | 76 | 51 |
| | Female | 74 | 49 |
| Educational Level | Primary | 31 | 20 |
| | Secondary | 34 | 23 |
| | Tertiary | 9 | 6 |
| | None | 76 | 51 |
| Occupation | Farming | 73 | 48 |
| | Fishing | 16 | 11 |
| | Trading | 39 | 26 |
| | Student | 12 | 8 |
| | Others | 10 | 7 |

Table 2 summarizes the knowledge and attitude of respondents on bioecology of *S. damnosum* complex. All the respondents 150 (100%) affirmed that they know black flies when probed of the presence of the fly in the community with its local or common name 'Amukuru' i.e causing itching. They unanimously established the fact that the flies bite seriously in the community. A larger percentage (73%) of the respondents established that the flies bite mostly at the riverside (Owena River), while 27% claimed they bite mostly at the farm. Information from the Focus Group Discussion (FGD) corroborated these observations. The flies were believed to be biting any part of the exposed body but with affinity for the leg (124) (82%) than any other

**Table 2. Knowledge and attitude of respondents on bioecology of *Simulium damnosum* complex.**

| Parameter | Response | No. of Respondent | Percentage |
|---|---|---|---|
| Have you heard of black flies? | Yes | 150 | 100 |
| | No | 0 | 0 |
| Do they bite in your community? | Yes | 150 | 100 |
| | No | 0 | 0 |
| Where do they bite in the community? | Farm | 38 | 25 |
| | Riverside | 109 | 73 |
| | House | 2 | 1 |
| | Others | 1 | 1 |
| Which part of the body do they often bite? | Head | 9 | 6 |
| | Neck | 13 | 9 |
| | Leg | 124 | 82 |
| | Others | 4 | 3 |
| What are the effects of their bite? | Itching | 121 | 80 |
| | Swelling | 28 | 19 |
| | Others | 1 | 1 |
| Where do they breed? | River | 46 | 31 |
| | Tree | 104 | 69 |
| | Others | 0 | 0 |
| Which season do they bite most? | Wet | 147 | 98 |
| | Dry | 3 | 2 |

**Table 3. Knowledge of respondents on onchocerciasis, causes and treatment.**

| Parameter | Response | No. of Respondent | Percentage |
|---|---|---|---|
| Do you know black flies cause onchocerciasis? | Yes | 18 | 12 |
| | No | 132 | 88 |
| If no, what causes onchocerciasis? | Witchcraft | 34 | 23 |
| | Hereditary | 5 | 3 |
| | Mosquitoe bite | 54 | 36 |
| | Eating kola nut | 16 | 11 |
| | No idea/uncertain | 41 | 27 |
| Are you aware of any medication for onchocerciasis? | Yes | 17 | 11 |
| | No | 133 | 89 |
| Is the medication orthodox or tradition? | Orthodox | 5 | 3 |
| | Traditional | 12 | 8 |
| Is onchodermatitis curable? | Yes | 3 | 2 |
| | No | 2 | 1 |
| | Uncertain | 145 | 97 |

part of the bodies. This is also affirmed during the FGD. The major effect of the fly's bite was attributed to itching by the respondents (80%) while 28(19%) respondents claimed it causes swelling.

About 69% of the respondents claimed that black flies breed in tree holes. Respondents during FGD also ascribed the breeding of black flies to tree holes such as 'Iroko kekere' and 'Igi araba". Most of the respondents confirmed that black flies bite more in the wet season 147 (98%) as against dry season 3(2%)

Table 3 shows that 88% of the respondents were ignorant of black flies as vectors of onchocerciasis. The level of ignorance was also evident among participants in the Focus Group Discussion (FGD). 36% believed onchocerciasis is caused as a result of mosquito bite, 23% attributed it to witchcraft, 11% claimed it's the consequence of eating kola nut, 3% see it as being hereditary while 27% have no idea of its cause.

A larger percentage (89%) of the respondents is unaware of any medication for the treatment of onchocerciasis while 11% are aware. However, few respondents claimed the use of orthodox medicine while 8% claimed it is traditional.

Table 4 summarizes the knowledge of respondents on methods of black flies bites prevention. There had hardly been any sensitization on onchocerciasis according to 89% of the respondents and later confirmed during the FGD. However, 91 (60%) of respondents prevent fly bite by clothing their bodies (such as wearing socks, long garments and sweater) while 52 (35%) rub ointment such as palm oil or mosquito repellent creams. Biting is usually intense in the morning according to 83% of the respondents, followed by evening (24) (16%) while fly bite in the afternoon is the least (1) (1%). After fly bites, majority 48% do nothing apart from killing the fly, 10% rub ointment, 7% go to the hospital while 4% take drugs (particularly anti-malarial drugs).

## Discussion

The result on perception of the community residents on bioecology of black flies showed that they are conversant with the black fly bites. A larger population of the residents affirmed that the flies bite mostly along the river side corroborating the scientific findings that people working close to the rivers are at the high risk of *Simulium* biting nuisance and onchocerciasis [15,16]. The reason for the impressive knowledge of the daily and seasonal distribution of the

**Table 4. Knowledge of respondents on methods of prevention of black flies bite.**

| Parameter | Response | No. of Respondent | Percentage |
|---|---|---|---|
| Has there been any sensitization on onchocerciasis? | Yes | 16 | 11 |
| | No | 134 | 89 |
| How do you prevent the flies from biting? | Clothing | 91 | 60 |
| | Rubbing of ointment | 52 | 35 |
| | Others | 7 | 5 |
| What time of the day is biting intense? | Morning | 125 | 83 |
| | Afternoon | 1 | 1 |
| | Evening | 24 | 16 |
| What do you do after fly bite? | Take drugs | 6 | 4 |
| | Go to hospital | 11 | 7 |
| | Rub ointment | 15 | 10 |
| | Nothing | 71 | 48 |
| | Others | 47 | 31 |

flies could be attributed to the long time presence of *Simulium* in the area. Furthermore, the river serves as a major source of water for domestic and occupational demand. Therefore reducing man-fly contact by refraining from such high biting areas of *Simulium* is paramount in controlling onchocerciasis.

The knowledge of the residents revealed that the flies bite mostly in the wet season than in the dry season which is in consonance with findings by [17]. This indicates that there is a higher risk of exposure to onchocerciasis in the wet season than in the dry season due to increase in adult fly population during the wet season. Thus, disease control is best carried out in the dry season since it gives a better chance for the destruction of the breeding site which invariably leads to the eradication of the flies.

Also, the poor knowledge of the residents on the breeding sites of *S. damnosum s.l* could increase their risk to onchocerciasis as majority of the residents do not have accurate knowledge of their breeding sites. This conforms to earlier reports on the poor knowledge of endemic communities on the ecology of *S. damnosum s.l* in many parts of Nigeria [18–20]. Majority of residents attributed the breeding site of black flies to trees and surrounding high vegetation. This misconception on the bio-ecology of the black flies could be as a result of the behavior of the flies in periods when the relative humidity is very low. Black flies are known to hide under shades and tree canopies [21]. Therefore, attributing the breeding sites of the vector to trees by residents may be due to their ignorance of the insect bio-ecology.

Majority of the respondents agreed that itching, pain and skin irritation are some of the observable effects of the bites of the fly. Skin disfiguration, lesions due to severe itching and scratching are some of the recognizable manifestations of onchocerciasis in many affected individuals [13,20].

Furthermore, 88% of the residents do not understand the aetiology of onchocerciasis as many assumed it is caused by mosquito bites. This assumption could be as a result of lack of sensitization and low level of awareness of the cause of the disease in the study area, this ignorance may lead to the persistence of onchocerciasis in the community. Public health enlightenment on black fly and onchocerciasis is therefore paramount for community members in a view to preventing the disease.

Since the residents lack knowledge of the cause of onchocerciasis, it is not unlikely their ignorance of the treatment for the disease as shown in the result. There is therefore the need for urgent sensitization on the treatment of the disease through the distribution of ivermectin

to residents who possibly show the disease symptoms. There is also the need for them to be acquainted not only with the available medications and modern tools for controlling human onchocerciasis but with an acceptable way of knowing the causative agent, mode of transmission and preventive measures. Visual demonstrations of parasite development in *Simulium* flies, transmission through their bites and identification of adult parasites in excised nodules may prove to be an effective tool in a health education programme.

Although, many reported to wear protective clothing when on the farms to prevent the flies from biting (p = 0.088; p>0.05). This mode of dressing in the period when farmers are preparing their farmlands for the next planting season is usually very uncomfortable, increases their body heat and retains sweats. Respondents in this study complained of the discomfort they endure while on the farm leading to them putting on cloths that exposes their lower legs and arms which exposes them to fly bite and consequently the risk of onchocerciasis. This is sequel to their claim that the fly bite do not prevent them from work or going out for their daily activities (p = 0.304; p>0.05).

The current study shows that the residents of the study community are unaware of the bioecology of blackflies and public health implication of onchocerciasis. Thus, necessitating the need for urgent intervention of the government at both the state and federal levels as well as public health organizations through effective community awareness with a view to eradicating the disease.

## Acknowledgments

The authors express their gratitude to the village head and residents of Alabameta community. Special thanks to the respondents for their cooperation towards the success of the study**.**

## Author Contributions

**Conceptualization:** Lateef O. Busari, Monsuru Adebayo Adeleke, Akeem A. Akindele, Olusola Ojurongbe.

**Investigation:** Lateef O. Busari.

**Supervision:** Monsuru Adebayo Adeleke, Akeem A. Akindele, Kamilu Ayo Fasasi, Olusola Ojurongbe.

**Validation:** Lateef O. Busari, Olabanji A. Surakat, Akeem A. Akindele.

**Writing – original draft:** Lateef O. Busari, Olabanji A. Surakat.

**Writing – review & editing:** Lateef O. Busari, Monsuru Adebayo Adeleke, Olabanji A. Surakat, Akeem A. Akindele, Kamilu Ayo Fasasi, Olusola Ojurongbe.

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
