## [Decision Letter · Decision Letter 0]

9 Jun 2021

Dear Mr Busari,

Thank you very much for submitting your manuscript "Black flies and Onchocerciasis: Knowledge, attitude and practices among inhabitants of Alabameta, Osun State, Southwestern, Nigeria" for consideration at PLOS Neglected Tropical Diseases. As with all papers reviewed by the journal, your manuscript was reviewed by members of the editorial board and by several independent reviewers. The reviewers appreciated the attention to an important topic. Based on the reviews, we are likely to accept this manuscript for publication, providing that you modify the manuscript according to the review recommendations. 

I thank the authors for submitting their work for our consideration. I agree with the reviewers that the manuscript addresses some novel aspects of onchocerciasis control efforts and should be published after minor revision. The reviewers identify a number of issues around the writing and presentation of the manuscript that require attention and improvement. I look forward to receipt of a suitably revised version of the paper and again thank the authors for submitting it to PLoS-NTDs

Sincerely,

Timothy G. Geary, PhD

Deputy Editor

Timothy Geary

Deputy Editor

I thank the authors for submitting their work for our consideration. I agree with the reviewers that the manuscript addresses some novel aspects of onchocerciasis control efforts and should be published after minor revision. The reviewers identify a number of issues around the writing and presentation of the manuscript that require attention and improvement. I look forward to receipt of a suitably revised version of the paper and again thank the authors for submitting it to PLoS-NTDs

Reviewer's Responses to Questions

**Key Review Criteria Required for Acceptance?**

**Methods**

-Are the objectives of the study clearly articulated with a clear testable hypothesis stated?

-Is the study design appropriate to address the stated objectives?

-Is the population clearly described and appropriate for the hypothesis being tested?

-Is the sample size sufficient to ensure adequate power to address the hypothesis being tested?

-Were correct statistical analysis used to support conclusions?

-Are there concerns about ethical or regulatory requirements being met?

Reviewer #1: 1st paragraph

-730 kilometer square = 730 km²

Reviewer #2: The background and rationale for the study are not well articulated. Therefore, although the study is to provide the knowledge attitudes and practices of the study population, it is difficult to know what the hypothesis was being tested. The statistics was to find difference in dynamics and transmission potential of blackflies. This is not possible through a questionnaire survey. One cannot determine transmission potential of vectors without undertaking vector studies.

**Results**

-Does the analysis presented match the analysis plan?

-Are the results clearly and completely presented?

-Are the figures (Tables, Images) of sufficient quality for clarity?

Reviewer #1: 1st paragr.

-what means “Respondents ….were higher than those…” ? You probably mean “were represented at a higher proportion…”

-delete “while students made up the rest of the respondents (8%)”

-is there actually a translation for “Amukuru”?

Reviewer #2: The study was supposed to have used some statistics in the analysis of the data, yet this is not seen in the results presented. One cannot therefore make any inferences on the analysis plan. The tables could be compressed to provide the key observations while the raw data could be added as an extra file

**Conclusions**

-Are the conclusions supported by the data presented?

-Are the limitations of analysis clearly described?

-Do the authors discuss how these data can be helpful to advance our understanding of the topic under study?

-Is public health relevance addressed?

Reviewer #1: (No Response)

Reviewer #2: The conclusions do not reflect on the data presented. The study's relevance does not come out clearly in the light of current onchocerciasis elimination goals. One key element of the study is not the knowledge on blackflies but the fact that there has not been any drug distribution in the community and the lack of knowledge of the treatment for the disease. The conclusion should advocate for either onchocerciasis elimination mapping in the area or the start of an alternate treatment strategy if the level of endemicity is already known. However, none of this is mentioned in the manuscript. Also there is less clarity in the report

**Editorial and Data Presentation Modifications?**

Reviewer #1: The acknowledgment is inappropriate. It should not thank any of the co-authors, nor family members or just anyone “who have contributed in one way or the other“. Only if there were specific contributions to be mentioned. On the other hand, was there any field staff to be mentioned here, and what about the community leaders/chiefs who certainly had to agree to the study in their village?

References:

Seem to be carelessly arranged! In the introduction, “WHO, 1994“ is cited, but not listed. Also, the references are not in strict alphabetical order, and different formats are used (journal abbreviated or not?; author initials with or without dots). Opara et al. is only cited for the 2005 paper, so no 2005a necessary.

Reviewer #2: I find the study of benefit to the elimination agenda for Nigeria. However, this is not demonstrated in the rationale for why the study was undertaken. The write up is also full of some mix ups. For example the authors indicate that the population is unaware of the bioecology of vectors while at the same time indicating that they are aware of the biting period in the day and season as well as which part of the body they prefer to bite. 

There should be a logical sequence in the write up starting from the abstract.

There should be clarity in data interpretation. For example at a point the authors indicate "Majority of residents attributed their breeding site to a tree “Iroko kekere” or “Igi araba and as such do not really pay attention to fly bite". Yet in another section indicate that the population wears protective clothing to prevent bites.

There is a need for a complete rewrite this manuscript but not necessarily getting new data

**Summary and General Comments**

Reviewer #1: The paper presents some rarely reported insights of local knowledge “at the end of the road” with regard to oncho and its vectors. Though easily ignored by the expert community, this knowledge can play a crucial role in oncho elimination efforts. Apart from some specific issues listed below, I would just like to point out the following for revision:

Since an APOC REMO in 2008 is mentioned, was there ever any oncho control interventions in the particular area? MDA? Can you give any prevalence data? Furthermore, this question should be addressed in the discussion. If people were never confronted with MDA or other oncho control activities, how should they know more about the disease? In any case, the conclusion still remains valid that health education needs to be implemented.

Reviewer #2: Overall i feel the study needs revision to be accepted for publication. There is a need for a completely re-haul of the write up to bring some clarity.

The strength of the manuscript is in the information it provides that there are still many areas with endemic onchocerciasis yet with not knowledge of elimination efforts. 

Unfortunately, the rationale for the study is not well articulated and the write up has many lapses of mixed up information. In the methodology section, one would expect to know what was done before the ethical approval is provided. The results section has no statistics for any of the observations made and finally, the discussion section is poorly written

There are many statements that needs clarity Examples are 

: "Alabameta is highly endemic with onchocerciasis with low community microfilarial load"

“It is endemic in many tropical countries”

“There were more male respondents (76) (51%) than female respondents (74) (49%).” Is this statistically different?

Please attached document.

PLOS authors have the option to publish the peer review history of their article (what does this mean?). If published, this will include your full peer review and any attached files.

Reviewer #1: No

Reviewer #2: Yes: Daniel Boakye

Figure Files:

Data Requirements:

Reproducibility:

References

---

## [Editor Report · Decision Letter 1]

22 Jan 2022

Dear Mr Busari,

Thank you very much for submitting your manuscript "Black flies and Onchocerciasis: Knowledge, attitude and practices among inhabitants of Alabameta, Osun State, Southwestern, Nigeria" for consideration at PLOS Neglected Tropical Diseases. As with all papers reviewed by the journal, your manuscript was reviewed by members of the editorial board and by several independent reviewers. The reviewers appreciated the attention to an important topic. Based on the reviews, we are likely to accept this manuscript for publication, providing that you modify the manuscript according to the review recommendations. 

I appreciate the work of the authors to respond to the concerns raised during the review process. The manuscript is much improved. However, a few important issues have not been adequately addressed; it will not be difficult for the authors to do so.

Please provide answers in the text to the following questions;

1. Why was this community chosen for the study? What kind of medical care service, if any, is available in the community? How many people live in the community? How were the 150 citizens chosen to participate in the survey?

2. Has the community been previously surveyed for the presence of onchocerciasis? If so, what was the prevalence? If not, why not? Presumably, ivermectin MDA has not been implemented there.

3. How close is the nearest community in which ivermectin MDA has been carried out? if MDA has not been carried out in the general vicinity, it may explain the lack of knowledge of the citizens. 

I again thank the authors for their efforts to improve the manuscript and look forward to a second revision, which should enable the paper to proceed to publication.

Sincerely,

Timothy G. Geary, PhD

Deputy Editor

Timothy Geary

Deputy Editor

I appreciate the work of the authors to respond to the concerns raised during the review process. The manuscript is much improved. However, a few important issues have not been adequately addressed; it will not be difficult for the authors to do so.

Please provide answers in the text to the following questions;

1. Why was this community chosen for the study? What kind of medical care service, if any, is available in the community? How many people live in the community? How were the 150 citizens chosen to participate in the survey?

2. Has the community been previously surveyed for the presence of onchocerciasis? If so, what was the prevalence? If not, why not? Presumably, ivermectin MDA has not been implemented there.

3. How close is the nearest community in which ivermectin MDA has been carried out? if MDA has not been carried out in the general vicinity, it may explain the lack of knowledge of the citizens. 

I again thank the authors for their efforts to improve the manuscript and look forward to a second revision, which should enable the paper to proceed to publication.

Figure Files:

Data Requirements:

Reproducibility:

References

---

## [Editor Report · Decision Letter 2]

10 Mar 2022

Dear Mr Busari,

We are pleased to inform you that your manuscript 'Black flies and Onchocerciasis: Knowledge, attitude and practices among inhabitants of Alabameta, Osun State, Southwestern, Nigeria' has been provisionally accepted for publication in PLOS Neglected Tropical Diseases.

Best regards,

Timothy G. Geary, PhD

Deputy Editor

Timothy Geary

Deputy Editor

---

## [Editor Report · Acceptance letter]

5 Apr 2022

Dear Mr Busari,

We are delighted to inform you that your manuscript, "Black flies and Onchocerciasis: Knowledge, attitude and practices among inhabitants of Alabameta, Osun State, Southwestern, Nigeria," has been formally accepted for publication in PLOS Neglected Tropical Diseases.

Best regards,

Shaden Kamhawi

co-Editor-in-Chief

Paul Brindley

co-Editor-in-Chief
